# Physico-Chemical and Sensory Characteristics of Extruded Cereal Composite Flour Porridge Enriched with House Crickets (*Acheta domesticus*)

**DOI:** 10.3390/foods14162893

**Published:** 2025-08-20

**Authors:** Tom Bbosa, Dorothy Nakimbugwe, Christophe Matthys, Jolien Devaere, Ann De Winne, Deniz Zeynel Gunes, Mik Van Der Borght

**Affiliations:** 1Research Group for Insect Production and Processing (IP&P), Department of Microbial and Molecular Systems, KU Leuven, Geel Campus, Kleinhoefstraat 4, 2440 Geel, Belgium; tom.bbosa@kuleuven.be; 2Department of Food Technology and Nutrition, Makerere University, Kampala P.O. Box 7062, Uganda; dnakimbugwe@gmail.com; 3Department of Chronic Diseases and Metabolism, Clinical and Experimental Endocrinology, KU Leuven, Gasthuisberg Campus, Herestraat 49, P.O. Box 7003, 3000 Leuven, Belgium; christophe.matthys@uzleuven.be; 4Department of Endocrinology, University Hospitals Leuven, Herestraat 49, P.O. Box 7003, 3000 Leuven, Belgium; 5Center of Aroma and Flavor Technology, Cluster Bioengineering Technology, Department of Microbial and Molecular Systems (M^2^S), KU Leuven, Campus Ghent, Gebroeders De Smetstraat 1, 9000 Ghent, Belgium; jolien.devaere@kuleuven.be (J.D.); ann.dewinne@kuleuven.be (A.D.W.); 6Department of Chemical Engineering, Soft Matter, Rheology and Technology (SMaRT), KU Leuven, Arenberg Campus, Celestijnenlaan 200 J, P.O. Box 2424, 3001 Leuven, Belgium; deniz.gunes@kuleuven.be; 7Department of Microbial and Molecular Systems, Center for Food and Microbial Technology (CFMT), KU Leuven, Arenberg Campus, Kasteelpark Arenberg 22, 3001 Leuven, Belgium

**Keywords:** edible insects, porridge acceptability, color parameters, aroma compounds, visco-elastic moduli, pasting properties

## Abstract

This study assessed the physico-chemical and sensory effects of enriching composite cereal porridges, typically consumed in Uganda, with undried house crickets (*Acheta domesticus*), a rich source of protein and vitamin B_12_. Composite flours containing 8.3% undried crickets, 66.7% maize, and 25.0% millet were compared to a control formulation with 73.0% maize and 27.0% millet, both extruded at 140 °C. Cricket enrichment slightly reduced lightness L* (59.99 vs. 61.28) and significantly increased aroma intensity (23,450 × 10^4^ AU vs. 18,210 × 10^4^ AU; *p* < 0.05), attributable to higher extrusion-induced Strecker degradation, Maillard reaction, and lipid oxidation. Rheological analysis revealed that paste made from cricket-enriched flour had lower critical strain (≈0.01%) and softened sooner than the control paste (≈0.03%) without becoming fragile. Both flours displayed stable paste-like behavior at stresses >10 Pa, with elastic moduli under 10^4^ Pa, which is typical for soft pastes. Reduced pasting values relative to native flours are attributable to starch pre-gelatinization during extrusion. Sensory evaluation showed positive hedonic ratings for both porridges, and a choice test indicated no significant consumer preference. Generally, physico-chemical and sensory changes were minimal, supporting the use of house crickets for nutrient enrichment of composite cereal porridges.

## 1. Introduction

Thin cereal porridges are a staple in Uganda as complementary foods for children and are commonly consumed by women during pregnancy and lactation [1]. However, the porridges have low nutrient density and poor bioavailability of nutrients [2], which could be improved by compositing with other nutritious ingredients, including edible insects, legumes like cowpeas, and wild fruits [3,4].

Consumption of edible insects in various processed forms is increasingly common. This could be fueled by several factors: curiosity, perceived health or nutritional benefits, sensory properties of insect-based foods, availability, or cultural appropriateness [5]. Incorporating edible insects as ingredients in other foods has been proposed by Henchion et al. [6] as a strategy to improve sensory acceptability and nutritional quality. Additionally, combination with favorite dishes is recommended to increase familiarity and consequently sensory acceptability of edible insects [7].

Rearing house crickets (*Acheta domesticus*) is increasingly being adopted in Uganda, mainly for their high nutritional content such as protein (60–71%), vitamin B_12_ (10–20 µg), iron (8.8 mg), zinc (19.6–20.2 mg), and calcium (150–171 mg) per 100 g of dry matter [2]. Accordingly, our team previously incorporated house crickets (either 8.3% undried or 4.2% dried forms) into millet and maize composite flours extruded at different temperatures (140–160 °C), leading to nutritional improvements. For instance, inclusion of undried crickets and extrusion at 140 °C delivered minimally higher contents of crude protein (12.63 vs. 10.77 g/100 g), ash (2.10 vs. 1.96 g/100 g), lysine (3.31 vs. 2.77 g/100 g protein), and valine (5.32 vs. 5.03 g/100 g protein) and a significantly higher amount of vitamin B_12_ (1.2 vs. 0.5 µg/100 g) than the control without house crickets on a dry matter basis [2]. To capitalize on the enhanced nutritional value of the process, it is essential that the resulting flour products exhibit both desirable physico-chemical properties and high sensory acceptability.

Sensory acceptability is a function of several attributes such as aroma, color, taste, and texture/rheological properties, which could be affected by addition of new ingredients, including house crickets. For instance, it was reported by Ayustaningwarno et al. [8] that house cricket inclusion in cookies leads to a darker and bluish cookie color. In addition, thermal processing likely alters the physico-chemical (color, aroma, pasting and rheological) and sensory characteristics, thus influencing overall sensory acceptability. For instance, the flavoring potential of insects is reportedly enhanced by thermal processing [9].

Pasting is the formation of a paste as starch granules swell and totally dislocate following gelatinization during heating and stirring of a starch/flour slurry [10]. Hence, pasting properties (Appendix A) reflect cooking properties of starch and can influence food texture and mouthfeel. However, pasting properties remain unexplored for cereal porridge containing house crickets. Additionally, the rheological and aroma properties of cricket-enriched porridges and flours remain unexplored.

While the inherent physico-chemical characteristics of flour samples have been studied, the overall human sensory profiles are paramount for food acceptability and preference. However, scant sensory evaluation studies [4,11] exist for cereal porridges containing house crickets. According to Aboge et al. [11] and Kinyuru et al. [4], cricket-containing porridges were less sensorially acceptable to women of reproductive age, care givers, and children, respectively. However, acceptability increased with the increase in exposure time, which relates to increased familiarity with house cricket taste and aroma. Notably, in only one instance [4] was extrusion processing applied to house-cricket-containing porridges. The house crickets used in the above study were oven-dried (50 °C, 72 h) and milled into cricket flour before co-extrusion (127 °C) with the other ingredients.

This study examines the physicochemical properties (color, aroma, pasting, and rheological characteristics) of extruded flour/porridge containing undried house crickets as well as the sensory properties of the porridge. This approach will contribute to a broader understanding of the acceptance of insect-based foods.

## 2. Materials and Methods

### 2.1. Sample Procurement and Preparation

Orange maize grain and millet flour were purchased from traders in Kampala district (Uganda) in March 2023. The flours and grain were packaged in polythene bags and stored at 25 °C. Maize grain was sorted and milled whole into flour using a hammer mill (SMECH, 9FQ series, Zibo, China). Adult house crickets (*A. domesticus*) were bought from a cricket rearing farmer in Masaka district (Uganda), transported in cool boxes, and stored at −20 °C before processing. Just before processing, the crickets were thawed to room temperature, washed until the wash water ran clear, and drained.

The formulation of materials/ingredients was carried out using the Linear Programming software, Nutrisurvey (version 16.0, 2007) (EBISpro, Willstätt-Legelshurst, Germany). The control flour was formulated by eliminating house crickets from the original formulation containing crickets. Two composite cereal flour formulations, one containing undried house crickets (140U) and a control without crickets (140C), were prepared. The flour sample, 140U, comprised 8.3% undried crickets, 66.7% maize, and 25.0% millet, while 140C comprised 73.0% maize and 27.0% millet.

### 2.2. Extrusion Processing

Both flours were extruded using a double-screw extruder (DP 70—III, Jinan Eagle Machine Co., Ltd., Jinan, China) at a feed rate of 0.4 kg/min and at a constant feed moisture content (adjusted to 14–16%), screw speed (150 rpm), and temperature (140 °C). The average initial feed moisture content (8%) was adjusted to 15% by adding 7.0 g of water/100 g of feed (flour) to improve extrudate texture.

The extruded pellets were milled using an industrial crusher (30B-C series universal crusher, Changzhou Yuanze Drying Equipment Co., Ltd., Changzhou, China) into flour samples and packaged in low-density polyethylene bags (Nile plastics, Giza, Egypt), stored at 25 °C in a dark room until further analysis and porridge preparation.

### 2.3. Color

Color analysis was carried out after three weeks of storing the samples using a colorimeter (Konica Minolta, Inc., Tokyo, Japan) in the CIE L*a*b* system. The color parameters L* (lightness), a* (redness/greenness), and b* (yellowness/blueness) were determined, and the browning index (BI) and change in chroma (ΔC) of the flour samples were calculated according to Ssepuuya et al. (Equations (1) and (2)) [12].
(1)$$\mathrm{BI}=100\;\frac{X-0.31}{0.17}\;\mathrm{where}\;X\;=\frac{a^{*}+1.75\;L^{*}}{5.645\;L^{*}+a^{*}-3.012\;b^{*}}$$
(2)$${\text{∆}C={\;C}_{\mathrm{abC}}-C_{\mathrm{abU}}=\left(a_{C}^{*2}+b_{C}^{*2}\right)}^{\frac{1}{2}}-{\;\left(a_{U}^{*2}+{\;b}_{U}^{*2}\right)}^{\frac{1}{2}}$$where C_ab_ is the chroma calculated using a* and b* values, while C and U denote the types of samples for which chroma is calculated.

The total color difference (ΔE) was calculated according to Fernández-Artigas et al. (Equation (3)) [13]:
(3)$${\text{∆}E\;=\;\left({\Delta L}^{*2}\;+{\Delta a}^{*2}\;+{\Delta b}^{*2}\right)}^{\frac{1}{2}}$$ where ΔE is the color difference between the flour containing house crickets and the control. ΔL* = lightness difference, Δa* = redness difference, and Δb* = yellowness difference.

### 2.4. Aroma Profile

#### 2.4.1. Sample Preparation

A sample (15 g ± 0.1%) of each composite flour was mixed with 100 mL of pre-boiled demineralized water maintained at 90 °C. The mixture was stirred until the water was completely absorbed by the flour and allowed to cool to a temperature of 50 °C. A (5 g ± 0.1%) portion of the warm mixture was transferred into a 20 mL headspace vial and hermetically sealed with an aluminum crimp cap lined with a PTFE/silicone septum.

#### 2.4.2. Analysis of Volatile Compounds

Volatile compounds were analyzed using headspace solid-phase micro extraction coupled with gas chromatography and mass spectrometry (HS-SPME-GC-MS) based on the method described by Pérez-Santaescolástica et al. [14], with modifications to optimize extraction from the porridge matrix. Analyses were performed on an Agilent 6890 gas chromatograph coupled with an Agilent 5973 mass-selective detector (Hewlett Packard, Palo Alto, CA, USA) using a Gerstel MPS2 autosampler (Gerstel GmbH & Co. KG, Mülheim an der Ruhr, Germany) equipped with a thermostatic agitator and fiber conditioning station. Samples were incubated for 10 min at 60 °C with intermittent agitation at 500 RPM (on for 5 s/off for 2 s). To avoid cross-contamination, a 50/30 µm divinylbenzene/carboxen/polydimethylsiloxane SPME fiber (Sigma Aldrich, Bornem, Belgium) was simultaneously conditioned for 5 min at 270 °C before exposure to the sample headspace for 25 min at 60 °C.

The extracted volatiles were subsequently separated using an Agilent J&W HP-INNOWAX capillary column (30 m × 0.25 mm × 0.25 µm) operating in constant flow mode using helium (99.99%) as a carrier gas (1 mL/min). The SPME fiber was desorbed in splitless mode for 5 min in the 0.75 µm ID SPME liner (Sigma-Aldrich, Bornem, Belgium) of the GC inlet at 250 °C. The following time–temperature program was applied: 5 min at 40 °C, followed by a temperature ramp at 4 °C/min to 240 °C, which was held for 5 min. The transfer line was maintained at 250 °C. The total ion current (70 eV) was recorded over the mass range from m/z 40 to 230 (scan mode) using no solvent delay and a threshold of 50 counts.

Volatile compound identification was performed based on two criteria: by comparing (1) the mass spectra of each compound with the Wiley 275 MS spectral library, and (2) compound-specific experimental Kovat’s Indices (KI_exp_) to those reported in the literature (KI_lit_) found through the NIST Chemistry WebBook online database [15]. Experimental Kovat’s Indices were determined using Van Den Dool and Kratz’s equation for temperature-programmed GC conditions and the retention times of a series of n-alkane homologues (C_8_–C_20_), which were analyzed using identical experimental conditions (Appendix A).

### 2.5. Pasting Properties of Flours

Pasting properties were determined according to Khatun et al. [16]. Reconstituted flours (1.8 g flour in 15 mL distilled water) were pasted in a controlled-stress rheometer (Anton Paar Physica MCR 301, Ostfildern, Germany), fitted with a starch pasting cell (C-ETD 160) comprising an aluminum measuring cup (CC26/ST) and a stirrer probe (ST 24-2D/2V/2V-30). Suspensions were first mixed for 10 s and held at 50 °C for 50 s, then heated from 50 to 95 °C at 0.1 °C/s, held at 95 °C for 5 min, followed by cooling from 95 to 50 °C and holding at 50 °C for 2 min. Parameters obtained were peak viscosity, breakdown viscosity, set-back viscosity, and pasting temperature, as defined by Appendix A, adopted from Balet et al. [10]

### 2.6. Rheological Properties of Porridges

Rheological measurements were determined according to Calabrese et al. [17] with some modifications as described below. For all rheological measurements, the sample was prepared by mixing 10 g of flour with 60 mL of boiling distilled water with constant stirring for 30 s. The prepared suspension was held in a 50 °C water bath with a shaker. A 22 mL sample was transferred into the rheometer cup (measuring cup radius 14.46 mm) and covered to reduce moisture loss. Temperature was controlled using a water bath connected to the Peltier plate of the rheometer.

Dynamic oscillatory measurements (amplitude/strain and frequency sweeps) were conducted using the rheometer (Anton Paar Physica MCR 301, Ostfildern, Germany) with a concentric cylinder geometry of 27 mm and a 40 mm measuring bob outer diameter and height, respectively, and a measuring gap size of 1.129 mm. Data for rheological measurements were analyzed with the rheometer software Rheoplus 32 Multi 6 version 3.40.

The strain sweep measurements were performed at a constant frequency of 1 Hz, temperature of 50 °C, and strain amplitude range from 0.01–100%. The frequency sweep measurements were performed at constant strain amplitude within a linear visco-elastic (LVE) range of 0.1% and frequency range of 0.1–100 Hz at a temperature of 50 °C. Furthermore, the rheological properties were measured as a function of the sample’s age with the goal of correcting for possible age-dependent properties, which could affect the amplitude sweep and frequency sweep data, in case of significant aging. To generate rheological data as a function of the samples’ age, samples were monitored at a steady frequency of 1 Hz and strain amplitude of 0.01% for 1 h.

### 2.7. Sensory Characteristics of Porridge

To make porridge for sensory tests, 50 g of flour was mixed with 300 mL of hot boiling distilled water (1:5), and 15 g of sugar was added. The flour was continuously stirred for 5 min, kept in a vacuum flask and served after cooling to 30 °C [18]. Porridge sensory acceptability and preference were determined using a structured questionnaire (Appendix A). A total of 60 semi-trained panelists (25 female and 35 male) at the Department of Food Technology and Nutrition, Makerere University (Uganda) undertook both sensory acceptability and preference tests [19]. To protect participants’ privacy and rights, written consent was sought after disclosure of potential risks and benefits of the sensory study. All participants testified to being regular consumers of cereal porridges, while the use of house crickets as an ingredient in porridge is increasingly being promoted. Additionally, study participants were free to withdraw from the study, and their data would only be shared for research purposes anonymously.

Approximately 15 mL three-digit blind-coded porridge samples in disposable cups were served to individual semi-trained sensory panelists in a well-ventilated sensory laboratory containing separate testing booths, with access to natural daylight conditions (room temperature 25 °C). All sensory analyses were conducted between 9:00 am and 12:30 pm. Panelists were asked to taste each sample independently and instructed to complete at least one third of each. They were also instructed to rinse their mouth before starting and in-between different samples using drinking water, which was provided.

For sensory acceptability, panelists were asked to rank each of the attributes using the provided 9-point hedonic scale, which describes the degree of liking/disliking. After the sensory acceptability test, panelists were asked about which of the two products they preferred (paired-preference test) during the same sitting. The design also included a no-preference option. Additionally, panelists were also asked to mention the most liked attribute about the preferred sample.

### 2.8. Data Analysis

All physico-chemical measurements were carried out in triplicate, and 60 semi-trained panelists participated in the sensory analysis. Sensory acceptability and physico-chemical measurement data were analyzed using the JMP Pro software, version 17.0 (JMP Statistical Discovery LLC, Cary, NC, USA), at a 5% significance level. Normality and homoscedasticity of all data sets were checked using Shapiro–Wilk’s and Levene’s tests, respectively. Data sets which met both the normality and homoscedasticity assumptions were analyzed using the paired t-test statistic. The Kruskal–Wallis test was used for sample comparisons in cases where the normality or homoscedastic assumptions were not met. Standard deviations of the calculated color values (BI, ΔC, and ΔE) were derived from the standard deviations of the measured quantities using the principles of error propagation as detailed by Farrance and Frenkel [20].

## 3. Results and Discussion

### 3.1. Color of the Composite Flours

Table 1 shows the color parameters of the flour containing undried house crickets (140U) and the control without crickets (140C), both extruded at 140 °C.

Flour sample 140U had a significantly lower value for lightness (L*) in comparison with 140C, suggesting darkening and a higher browning index (BI) for 140C (*p* < 0.05). Although the measured color difference (ΔE = 1.30) falls below 2, the threshold of perceptibility for the average observer [13], the panelists observed a slightly darker appearance for the porridge containing undried house crickets (made from flour 140U). The levels of redness/greenness (a*) and yellowness/blueness (b*) were statistically similar for both studied flours. Flour images are also represented by Appendix A to demonstrate the observed flour visual perception.

For 140U, pre-extrusion enzymatic browning could have occurred [21,22] along with temperature induced non-enzymatic browning (Maillard reactions) [12] during extrusion. Besides already existing phenolic compounds in maize and millet flours, maceration of undried house crickets during composite flour formulation could have exposed phenolic compounds contained in the crickets to enzyme-catalyzed oxidative browning [21,22]. Additionally, the exposed phenolic compounds could have further reacted with proteins and amino acids, forming dark-colored melanins [22]. This could explain the slightly darker appearance of the porridge from 140U (sensory panelist observations). The slight color changes could affect sensory acceptability of the products if the amount of undried house crickets added is increased.

Notably, the type of packaging material, storage temperature, and period are likely to influence flour color changes due to both enzymatic and non-enzymatic browning reactions. Therefore, future research should focus on the effect of storage conditions on the color of extruded cricket containing cereal flours.

### 3.2. Aroma Profile of the Composite Flours

The peak areas (expressed as AU × 10^4^/g) of different classes and individual aroma compounds and their contribution to the aroma of the studied composite flours are presented in Table 2.

It worth noting that the aroma results require cautious interpretation, since the units of peak areas may not be directly proportional to the actual aroma concentrations of the compounds studied, and the sensory contribution of some aromatic compounds is not yet known. A graphical representation of the different aroma classes is included in Appendix A.

The aroma compounds identified include aldehydes, furan derivatives, benzene derivatives, alcohols, ketones, pyrazines, pyrroles, hydrocarbons, terpenes, lactones, sulfur compounds, and acids. Significantly higher total peak areas of branched aldehydes (*p* = 0.01), aromatic aldehydes (*p* = 0.03), ketones (*p* < 0.01), pyrazines and pyrroles (*p* < 0.01), hydrocarbons (*p* < 0.01), terpenes (*p* = 0.05), and sulfur compounds (*p* < 0.01) occurred in 140U compared to 140C (*p* ≤ 0.05). Esters were not found in the flour samples, possibly due to degradation during extrusion cooking, since esters are heat-sensitive, while up to 16 different esters have previously been found in both oven- and freeze-dried house crickets [26].

Among aldehydes, saturated aldehydes represent the highest peak area, followed by aromatic, unsaturated, and branched aldehydes. Among saturated aldehydes, heptanal had a significantly higher peak area in 140U than 140C (*p* < 0.01), while the peak area for octanal was significantly higher for 140C (*p* = 0.03). Heptanal has been previously identified by Umebara et al. [37] in house cricket powder from crickets fed on apple by-products. Consistent with our findings, unheated house crickets have been previously found to contain heptanal [14]. Since linear organic aldehydes are majorly derived from lipid oxidation [30], 140U possibly had higher amounts of unsaturated fatty acids, which are prone to lipid oxidation, than 140C.

The peak areas of all the individual aromatic aldehydes (benzaldehyde, safranal, and phenylacetaldehyde) were significantly higher in 140U relative to 140C (*p* = 0.03). Safranal has not previously been associated with edible insects, and it is a thermal degradation product of zeaxanthin [38], which is one of the carotenoids found in yellow maize and millet [39]. However, the feed-dependent presence of carotenoids in house crickets [40] is also likely. The occurrence of phenylacetaldehyde has previously been reported in oven-dried and blanched house crickets due to Strecker degradation of phenylalanine [26,41], while benzaldehyde (commonly reported aroma in edible insects) is a Maillard reaction product mainly formed from the Strecker degradation precursor, phenylacetaldehyde, and is responsible for off-flavors [30,42]. Despite the high peak areas, it is uncertain whether the benzaldehyde content in the flour samples is capable of meeting its high odor detection threshold to contribute to the aroma of the flour samples, while phenylacetaldehyde most likely played a major role.

As for unsaturated aldehydes, only the peak area of 4-heptenal was significantly higher in 140U than 140C (*p* < 0.01). Although the odor detection threshold of 4-heptenal is 0.025 µg/L water [23], it is uncertain whether 4-heptenal could have had an impact on the overall aroma of the products due to the relatively low peak area.

Three branched aldehydes (2-methylpropanal, 2-methylbutanal, and 3-methylbutanal) were found to be significantly higher in 140U compared to 140C (*p* ≤ 0.05). They are reportedly among the most reported branched aldehydes found in edible insects [30]. The aromas of 2-methylbutanal and 3-methylbutanal result from Strecker degradation of isoleucine and leucine, respectively, leading to an array of aromas such as buttery, toasty, fatty, floral, sweet, or roasted [41]. The odor detection thresholds of the observed branched aldehydes are generally low, reportedly 1.5 and 0.5 µg/L water for 2-methylbutanal and 3-methylbutanal, respectively [23,34]. Hence, the combination of large peak areas and low odor detection thresholds could significantly contribute to food aromas.

Six furan derivatives were identified, with four (3-methylfuran, 2-methylfuran, 2-ethylfuran, and 2-pentylfuran) having significantly higher peak areas in 140U than 140C (*p* ≤ 0.05). The most reported furan derivative in edible insects is 2-pentylfuran [30], which also has the highest peak area in our study. The low odor detection threshold of 2-pentylfuran (6 µg/kg) [24,32] and the high peak area may have contributed to the aroma of the flour products characterized by fruity, green, earthy, beany, buttery, fishy, and grassy flavor notes [30]. Peak areas of 2-ethylfuran are also relatively high; however, its odor threshold and aroma description are still unknown to the best of our knowledge. Both 2-methylfuran and 3-methylfuran have low peak areas, although they are significantly higher in 140U (*p* ≤ 0.05). Khatun et al. [26] identified 2-methylfuran in oven- and freeze-dried house crickets, contributing to pleasant aromas such as sweet, green, and fruity.

Oxidation of n-3 fatty acids and n-6 fatty acids results in formation of 2-ethylfuran and 2-pentylfuran, respectively [43], while 3-methylfuran could be derived from carotenoids in the ingredients used including house crickets [44]. Linoleic acid oxidation, oxidation of other n-6 fatty acids, and Maillard reactions are responsible for the formation of 2-methylfuran [45], while feed-dependent accumulation of carotenoids (ß-carotene, vitamin A, lutein, and zeaxanthin) has been reported in late-instar house cricket nymphs [40].

Benzene derivatives represent the third most common group in this study. They have previously been identified in thermally processed cereal products, including yellow maize flour and porridge, due to the degradation of carotenoids [39]. Their low odor detection thresholds could contribute to characteristic aromas. However, only 1,2,3-trimethylbenzene was found to be significantly higher in 140U as compared to 140C (*p* = 0.01) but its relative content, as shown by the peak area, is likely insufficient to influence flavor. The latter compound has also been previously identified in house cricket powder from crickets fed on apple by-products [37].

Generally, both samples had statistically similar peak areas (*p* > 0.05) for total alcohols, with total aliphatic alcohols generally higher than total phenols. The peak area of phenol was significantly higher in 140U than 140C (*p* < 0.01). Phenol exhibits tarry flavor notes, and its low aroma threshold is likely to contribute to the flour aroma [14,28]. Perez-Santaescolastica et al. [14] detected phenol in raw *Alphitobius diaperinus*, *Blaptica dubia*, *Tenebrio molitor*, and *Zophobas morio* but not in *A. domesticus*, which implies that heat processing could be responsible for phenol formation. The peak area of 1-hexanol was higher in 140C compared to 140U (*p* < 0.01). Notably, 1-hexanol has not been identified in house crickets by previous authors; however, it was identified by Ekpa et al. [39] in maize flour products.

Generally, the peak area of ketones was higher in 140U than 140C. Up to nine ketones were identified, six of which (2-propanone, 2,3-pentanedione, 2-heptanone, 2-octanone, 6-methyl-5-heptane-2-one, and 2-nonanone) were significantly higher in 140U as compared to 140C (*p* ≤ 0.05). Ketones are mainly formed from cleavage of lipid hydroperoxides and contribute to food flavors [26]; for example, 2-heptanone (highest peak area) formed from oxidation of saturated fatty acids has a strong cheesy and fruity odor [31]. Sample 140U possibly contained more fatty acids involved in lipid hydroperoxide cleavage reactions since it was previously found to contain a higher crude fat content than 140C [2]. Despite the generally low odor thresholds of ketones, their peak areas were small, and they hence were unlikely to influence overall flavor.

Pyrazines, characterized by toasted, roasted, nutty, cocoa-like, or coffee-like flavor notes are reportedly typical Maillard reaction products [46,47], as is the case for pyrroles [48]. The peak area of all six pyrazines and one pyrrole derivative in 140U were significantly higher than that of 140C (*p* ≤ 0.05). In thermally processed house crickets, 2,5-dimethylpyrazine and 2,5-dimethyl-3-ethylpyrazine were previously identified [26,46]. A generally higher amino acid content in 140U [2] may have contributed to formation of more pyrazines and pyrroles, with threonine and serine typically involved in pyrazine formation [48] while proline, is the likely source of pyrroles [49].

Pyrazines generally play a role as flavor and aroma enhancers and are reported to inhibit lipid oxidation [26]. Our findings indicate a higher number of methyl-substituted pyrazines than the ethyl-substituted ones. Methyl-substituted pyrazines reportedly have higher odor thresholds than ethyl-substituted ones [35]. The individual contribution of each pyrazine may be small, given the low peak surface area of each pyrazine, but the significantly higher peak surface area (*p* ≤ 0.05) of all pyrazines together may contribute specifically to the total aroma of 140U.

Four aliphatic hydrocarbon compounds were identified, with only 3-ethyl-2-methyl-1,3-hexadiene being significantly higher in 140U than 140C (*p* = 0.05), and its peak area units were also much higher than the rest of the identified hydrocarbons. Previous research on aroma compounds did not find 3-ethyl-2-methyl-1,3-hexadiene in house crickets. It has been reported that alkenes with two or more double bonds and methyl-branched alkenes are less common in other insects [50]. Hydrocarbons, generally formed by lipid oxidation and Maillard reactions, have little effect on flavor due to their high odor threshold, but some of them are reportedly important precursors of heterocyclic compounds during thermal treatment [30,51]. Additionally, hydrocarbons naturally occur in edible insects, including house crickets, as glandular secretions in alarm situations or cuticular compounds involved in species recognition such as during mating [30].

The terpene content was significantly higher in 140U than 140C (*p* ≤ 0.05). This can be explained by the occurrence of sesquiterpenes (α-zingiberene, b-bisabolene, b-sesquiphellandrene), which were exclusively found in 140U, and ar-curcumene, which was significantly higher in 140U than 140C (*p* < 0.01). While sesquiterpenes are known to be produced by plants as defense mechanisms, it has been confirmed that certain compounds occur in both plants and some insect species although not in insects of the order Orthoptera [52]. It has been hypothesized that either there has been an independent evolution of the biosynthetic pathways or a possible horizontal gene transfer from the plant to the animal kingdom, or possibly insects are accumulating the shared compounds from plant sources [52,53].

The house crickets in this study were reared on different mixtures containing maize bran and fish meal combined with different vegetables, including pumpkin leaves, moringa leaves, and either cassava or sweet potato leaves. Relatedly, α-zingiberene, b-bisabolene, and b-sesquiphellandrene are listed among the herbivore-induced sesquiterpenes produced by the maize plant as a defense mechanism [54]; however, ar-curcumene is not mentioned. Hence, it is possible that the plant-based feeds on which the house crickets were reared contained sesquiterpenes.

Only two sulfur compounds were identified, with 140U having significantly higher amounts of dimethyl sulfide (*p* = 0.02) and dimethyl disulfide (*p* = 0.05) than 140C. Both compounds are typical Strecker degradation products involving the amino acids cysteine and methionine during thermal processing [55]. Polysulfides such as dimethyl disulfide can also be formed from oxidation of methanethiol, a demethiolation product of sulfur containing amino acids, mainly cysteine and methionine [56]. Additionally, methanethiol can be formed by action of microbial enzymes on methionine and cysteine; hence, dimethyl disulfide could be a potential indicator of microbial spoilage in case any spores left after thermal processing germinate into vegetative forms during storage [30,56]. The odor detection thresholds of dimethyl sulfide and dimethyl disulfide [29] are very low. Hence, although the peak areas observed are low, sulfur compounds could have a small impact on aroma.

To conclude, inclusion of house crickets generally led to higher peak areas of most aroma compounds in 140U as compared to 140C. However, the real impact of each aroma compound can only be determined by comparing the flavor detection threshold and the concentration of each compound. Additionally, individual aroma compounds do not single-handedly affect the overall aroma of a food product. Therefore, in addition to compound interactions, minor components (due to additive effects) must also be considered. Further research is required on higher levels of undried house cricket inclusion (above 8.3%) in addition to establishing actual concentrations of aroma compounds and conducting in vivo aroma release studies.

### 3.3. Pasting Properties of Flour

The pasting profiles of 140U and 140C, represented by Figure 1, generally indicate that house cricket addition did not significantly affect pasting properties (*p* > 0.05).

All pasting properties were lower than what would be expected for native unextruded flours, demonstrating either partial or complete gelatinization of flour starch. For instance, typical pasting curves for native finger millet and extruded finger millet indicate a peak viscosity of 3000 mPa.s and > 500 mPa.s, respectively [57]. The viscous behavior of flours during cooking is a function of swelling potential and rigidity of starch granules and the extent of leaching of amylose into the solution [58,59]. It is generally determined by the starch, lipid, and protein content and the ratio of amylose to amylopectin present in the flours [10]. Viscosity analysis enables rapid determination of the cooking attributes of flours, which are generally affected by the extent of modification of starch granules during thermal processing [58].

Peak viscosity is the highest viscosity obtained during heating or pasting of a starch slurry. It represents the degree of gelatinization and water-holding potential of starch granules before physical breakdown [10]. The peak viscosity of 140C was higher than the value observed for 140U. According to Bbosa et al. [2], the control flour (140C) had a significantly higher carbohydrate content than 140U; hence, it is also likely to have a higher starch content. This was attributed to the substitution effect of house crickets in 140U, which are not known to contribute to carbohydrates. A high starch content is positively correlated with a high peak viscosity, while high fiber and protein contents negatively affect the peak viscosity of high-temperature-treated cereal flours [60]. However, the values obtained for both flours (141 and 155 mPa.s) in this study were lower than those reported for extruded maize flour (120 °C, 321.6 mPa.s) and non-extruded maize flour (1237 mPa.s) [58]. This could be attributed to higher starch solubility caused by starch degradation or dextrinization during extrusion [61,62].

Flour 140C had slightly higher values of both trough (holding strength) and breakdown viscosity than 140U. Trough viscosity, the minimum viscosity attainable after peak viscosity, results from destruction of starch granules, while the difference between trough (holding strength) and peak viscosity is the breakdown viscosity [10]. Hence, 140C forms a slightly more stable paste than 140U.

Set-back viscosity measures the tendency of starch to undergo association and retrogradation on cooling [62]. Set-back viscosity did not vary significantly between the two flour samples. However, the obtained values of 63.4 mPa.s for 140U and 72.9 mPa.s for 140C were higher than 44.33 mPa.s reported for native corn starch [63]. The relatively high set-back viscosity values are attributable to varying degrees of chemical bond formation between starch and non-starch components during extrusion cooking.

Final viscosity reflects paste- or gel-forming ability of the flour upon cooking or cooling and the extent to which the paste can withstand shear stress during stirring [64]. The studied flour samples have equal capabilities of forming a stable paste during cooking. Final viscosity (77.9 mPa.s) is lower than values reported for unextruded corn flour (3260 mPa.s) and extruded corn flour (120 °C) (342 mPa.s) [58].

The pasting temperature (Figure 1) is the minimum temperature required for the starch to cook/gelatinize [62]. Higher pasting temperature indicates higher resistance of starch granules to swelling and rupture. The pasting temperatures of the studied flours (62 and 57 °C) are lower than 92 °C reported for native corn starch [63], attributable to pre-cooking during extrusion.

Peak time (Figure 1) represents the number of minutes required by the flour to achieve maximum viscosity; hence, peak times indicate the ability of the flour to cook faster [10,64]. The peak time of the studied flour samples (5.18 s and 5.19 s) was lower than 5.73 s reported for native maize starch [63]. This can be attributed to pre-cooking during the process of extrusion for both flours.

Generally, inclusion of undried house crickets did not have significant impact on the pasting profile of extruded composite cereal flours.

### 3.4. Rheological Properties of Porridge

Rheological properties play a very important role in the transportation, delivery, and processing of food products and people’s taste satisfactions [65]. The results of small-amplitude oscillatory measurements for extruded composite cereal porridges with and without house cricket inclusion are presented in the proceeding sections.

#### 3.4.1. Aging Characteristics

Colloidal gel systems are reportedly in an out-of-equilibrium state; hence, they very often display time-dependent properties due to the tendency towards network restructuring (aging phenomenon) [66]. It was therefore legitimate to assume that some time evolution of the rheological properties could be observed versus the age of the samples (i.e., the measurement time after loading into the rheometer). Appendix A represents the visco-elastic moduli (elastic/storage modulus, G′, and viscous/loss modulus, G″) dependency on sample age.

The characterization of food material rheological aging is required to accurately report, control, and predict product stability, shelf life, and safety [67]. Generally, the porridge samples did not experience rheological aging, as evidenced by the generally constant curves of storage modulus (G′) and loss modulus (G″). Therefore, the frequency dependance of the moduli (Section 3.4.3) could be accurately determined without correcting for aging phenomena.

#### 3.4.2. Variation of Visco-Elastic Moduli with Amplitude Strain and Yielding Behavior

Figure 2 represents the visco-elastic moduli (elastic/storage modulus, G′, and viscous/loss modulus, G″) dependency on strain amplitude, while the yielding behavior of the flour samples is represented by Figure 3.

The studied flours displayed flow curves with well-defined linear regimes at low shear strain amplitude followed by strain-thinning properties, as expected for a paste consisting of a jammed system of particles. The linear visco-elastic region (LVE) was evidenced by constant G′ and G″ values until the critical strain (γ_c_) was achieved, measured at ≈0.01 and ≈0.03% for 140U and 140C, respectively. The critical strain (γ_c_, the point at which the change in G′ exceeds 5%) represents a counterbalance between maintaining gel structural integrity and shear deformation of the sample [68,69]. The lower critical strain (≈0.01%) of 140U compared to 140C (≈0.03%) suggests that the paste with crickets started to soften sooner, however without becoming fragile.

The absence of fragile behavior of the paste with crickets can be inferred from the data recorded at higher values of strain amplitude. When increasing the strain amplitude significantly above the critical strain, the decrease in G′ was higher than that of G″ for both samples, indicating first a softening until G′ and G″ curves cross, beyond which the system’s response is then dominated by viscous dissipation rather than an elastic response. The cross-over point is in some publications identified as the yield point; however, that is unsuitable for concentrated dispersions, for the G′ and G″ curves do not always cross [17]. Sample 140C easily undergoes softening and yielding at lower strain values and stress values (cross-over at strain γX_c_ = 0.30% (Figure 2), yield stress *σ*_y_C = 1.12 Pa (Figure 3)) compared to 140U (cross-over strain γX_u_ = 4.23% (Figure 2), yield stress *σ*_y_U = 3.99 Pa (Figure 3)). The yield point for the 140C system was found just before the cross-over point; hence, the cross-over strain and yield strains took close values, unlike the 140U system, where the cross-over strain was much higher than the yield strain. This indicates that porridge from 140C has a crumblier texture than that of 140U. House crickets contain chitin, which has been reported to increase both G′ and G″ through its colloidal interaction involving hydrogen bonding [70,71]. Similarly, addition of 5–15% house cricket flour in wheat flour was found to increase both G′ and G″ [16]. Noteworthy is the fact that these systems are very stable to flow-induced permanent damage under the effect of moderate stresses, i.e., lower than 10 Pa, as under the protocols used for rheological characterization.

#### 3.4.3. Angular Frequency Dependence of Visco-Elastic Moduli

The variation in the visco-elastic moduli with angular frequency is represented by Figure 4.

The studied porridges displayed flow curves with a storage modulus (G’) greater than the loss modulus (G″) without crossing-over of the moduli over the entire frequency range, which indicates paste-like behavior; however, they are categorized as soft pastes since the elastic moduli are always under 10^4^ Pa. A slight frequency dependency was observed along the frequency range studied, indicating very little relaxation at the experimental time scales explored.

### 3.5. Sensory Attributes and Preference of Porridges

The hedonic rating scores of the flours for the different sensory attributes are presented in Figure 5. The results indicate that the two porridges generally had hedonic scores above 5, which depicts that both were liked.

Generally, the hedonic scores of the studied porridge sensory attributes are statistically similar. This implies that the 140U and 140C porridges were equally accepted. The slightly higher acceptance of the 140U porridge than 140C porridge can be attributed to the observed significantly higher area units of aroma compounds in the 140U flour. As a result, 140U is likely to compete favorably with pre-existing flour products without house crickets in terms of consumer preferences. Panelists’ preference characteristics are presented in Figure 6.

The results show a slight preference of the panelists for 140U than 140C. Taste and aroma are the main drivers of choice for the studied porridge products. While we have identified taste and aroma as the main drivers of choice for both the 140U and 140C porridges, other extrinsic factors such as information about health benefits, neophobic tendencies, and willingness to pay [7,72] could determine repeated trials to consume edible insects or foods containing edible insects. Nevertheless, participants’ familiarity with cereal porridges as a staple food in Uganda [1] could have played an important role in the acceptability of porridge containing house crickets.

## 4. Conclusions

This study compared the physico-chemical and sensory characteristics of extruded composite maize/millet flour with (140U) and without undried house crickets (140C) both extruded at 140 °C. Incorporating undried house crickets (8.3%) significantly increased aroma peak areas attributable to a higher occurrence of typical extrusion-induced Strecker degradation and Maillard and lipid oxidation reactions. The color, pasting, and rheological characteristics were generally maintained, with 140U being slightly browner and more rheologically stable than 140C. Most importantly, the sensory properties of the porridges remained unaltered while improving the nutrient quality of the flour by incorporating undried house crickets. Future research should consider establishing actual concentrations of aroma compounds and thresholds rather than using peak areas in addition to in vitro odor release studies. Furthermore, a higher cricket inclusion (more than 8.3%) should be explored, as well as investigating the effect of storage and packaging conditions on physico-chemical properties.

## Figures and Tables

**Figure 1 foods-14-02893-f001:**
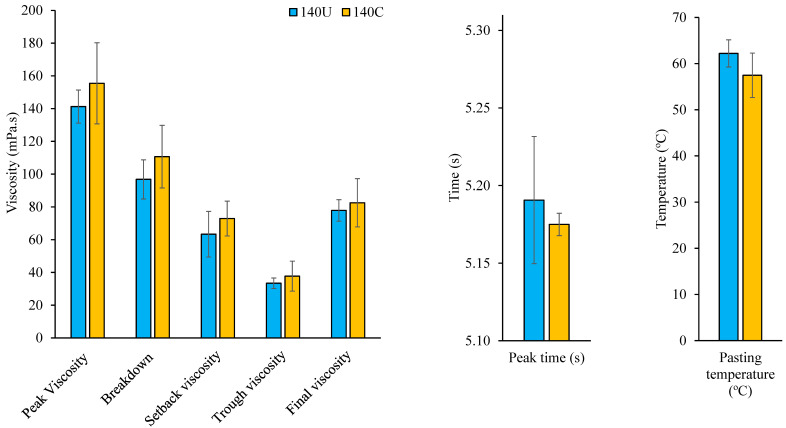
Pasting properties of flours with (140U) and without (140C) undried house crickets both extruded at 140 °C (*n* = 3). The error bars indicate standard deviations. Statistical differences were not observed based on *p* ≤ 0.05.

**Figure 2 foods-14-02893-f002:**
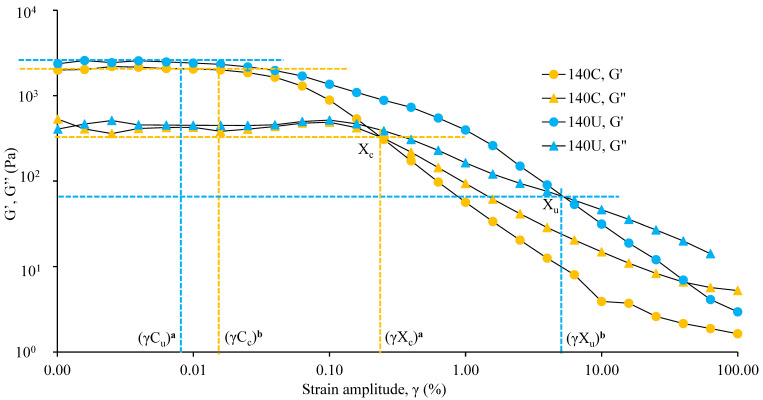
Visco-elastic moduli (elastic/storage modulus, G′, and viscous/loss modulus, G″) as a function of strain amplitude, γ, for porridges containing undried house crickets (140U) and the control without house crickets (140C) (*n* = 3). γC_u_ represents the critical strain for sample 140U = 0.01% obtained at G′ = 2685 Pa, γC_c_ represents the critical strain for sample 140C = 0.03% obtained at G′ = 1711 Pa. γX_c_ (0.30%), and γX_u_ (4.23%) represents the strain amplitudes at cross-over points X_c_ and X_u_, respectively. Different superscripts (bold) indicate significant differences for critical strain and strain at cross-over (*p* ≤ 0.05).

**Figure 3 foods-14-02893-f003:**
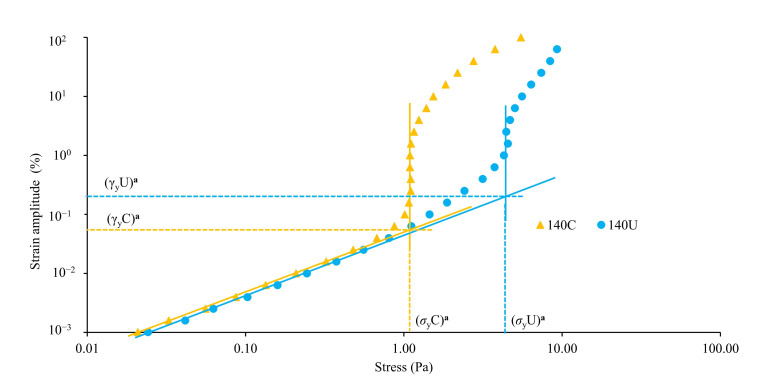
Strain amplitude (%) as a function of stress (Pa) for porridges containing undried house crickets (140U) and the control without house crickets (140C) (*n* = 3). Yield strain: γ_y_U = 0.21%, γ_y_C = 0.07% for 140U and 140C, respectively. Yield stress: σ_y_U = 3.99 Pa, σ_y_C = 1.12 Pa for 140U and 140C, respectively. Similar superscripts in bold indicate no significant differences (*p* ≤ 0.05).

**Figure 4 foods-14-02893-f004:**
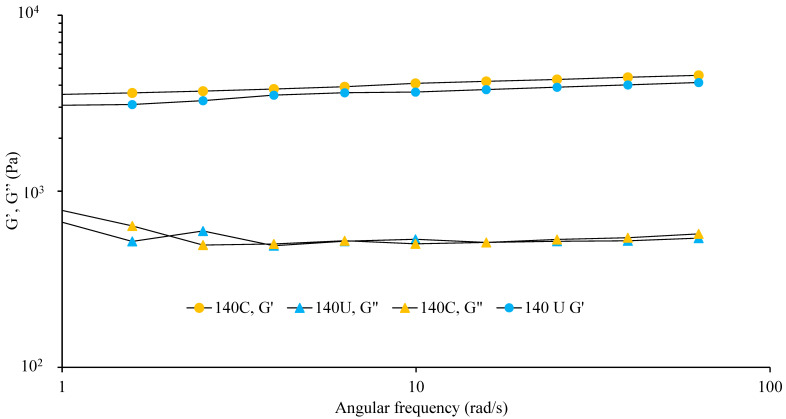
Visco-elastic moduli (elastic/storage modulus, G′, and viscous/loss modulus, G″) as a function of angular frequency, rad/s, for porridges containing undried house crickets (140U) and the control without house crickets (140C) (*n* = 3).

**Figure 5 foods-14-02893-f005:**
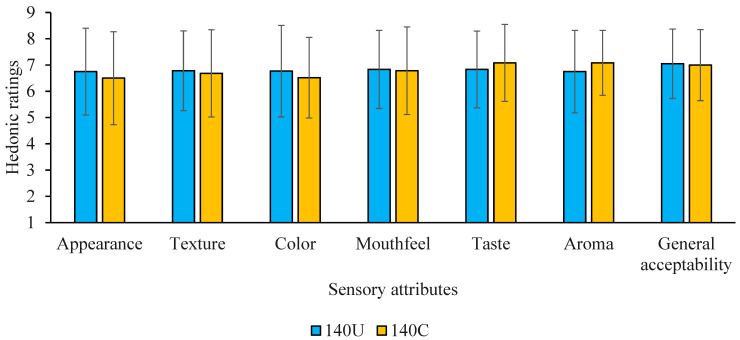
Sensory acceptability profile of a semi-trained sensory panel (*n* = 60) for extruded composite cereal porridges containing undried house crickets (140U) and the control without crickets (140C). The error bars indicate standard deviations. Statistical differences were not observed based on *p* ≤ 0.05. The *x*-axis represents sensory attributes. The *y*-axis represents ratings based on the 9-point hedonic scale: 1 = dislike extremely, 2 = dislike very much, 3 = dislike moderately, 4 = dislike slightly, 5 = neither like nor dislike, 6 = like slightly, 7 = like moderately, 8 = like very much, 9 = like extremely.

**Figure 6 foods-14-02893-f006:**
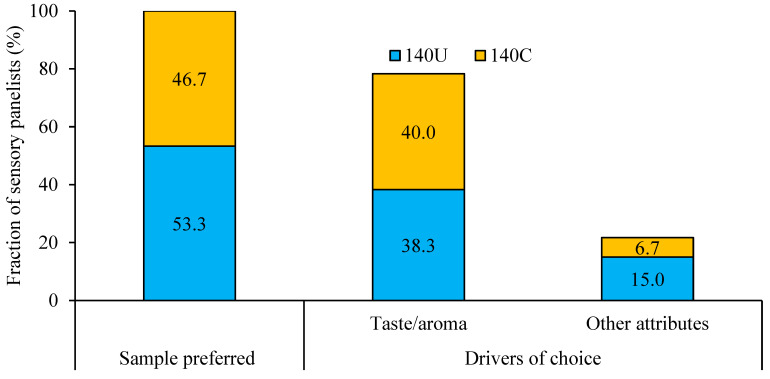
Preference characteristics of porridges with undried house crickets (140U) and control without house crickets (140C) (*n* = 60).

**Table 1 foods-14-02893-t001:** Color properties of composite flour samples with and without house crickets extruded at 140 °C (*n* = 3). The values are represented as means ± standard deviation.

Color Parameter	Sample Treatments	
	140U	140C
Lightness (L*)	59.99 ± 0.13 ^a^	61.28 ± 0.33 ^b^
Redness (a*)	6.31 ± 0.00 ^a^	6.24 ± 0.06 ^a^
Yellowness (b*)	22.90 ± 0.09 ^a^	23.01 ± 0.19 ^a^
Browning index (BI)	54.85 ± 0.01 ^a^	53.62 ± 0.02 ^a^
Chroma (C)	23.75 ± 0.09 ^a^	23.84 ± 0.19 ^a^
ΔC	−0.09 ± 0.20	
ΔE	1.30 ± 0.35	

Values with different superscripts along the column for each parameter are significantly different (*p* < 0.05). ΔC denotes chroma difference between the two samples, ΔE denotes total color difference.

**Table 2 foods-14-02893-t002:** Volatile compounds detected in composite flour samples with and without house crickets extruded at 140 °C expressed as AU × 10^4^/g (*n* = 3). The peak areas of aroma compounds are presented as means ± standard deviations. Different superscripts along the rows indicate significant differences between 140U and 140C (*p* ≤ 0.05).

Compound	140U	140C	Sensory Attribute	Recognition Thresholds
	Mean ± StDev			
Total aldehydes	8067 ± 332 ^a^	7042 ± 760 ^a^		
*Total saturated aldehydes*	*6082 ± 273 ^a^*	*5489 ± 454 ^a^*		
Hexanal	4088 ± 228 ^a^	4021 ± 384 ^a^	Green, apple, grassy, aldehydic, fresh, fruit, oil	10 µg/L
Heptanal	1092 ± 95 ^a^	495 ± 41 ^b^	Citrus, fat, green, nut, floral, dry fish	0.01 mg/kg
Nonanal	353 ± 75 ^a^	423 ± 49 ^a^	Citrus, fatty, green, aldehydic	8 µg/L
Pentanal	337 ± 49 ^a^	334 ± 28 ^a^		0.008 mg/kg
Octanal	138 ± 16 ^b^	184 ± 18 ^a^	Citrus, grassy, green, fat, soap, lemon, mushroom, moldy	6.9 µg/L
*Total aromatic aldehydes*	*1185 ± 82 ^a^*	*880 ± 137 ^b^*		
Phenylacetaldehyde	623 ± 79 ^a^	401 ± 97 ^b^	Berry, geranium, honey, nut, pungent	5.2 µg/L
Benzaldehyde	544 ± 2 ^a^	468 ± 37 ^b^	Fruity, sweet, bitter almond, burnt sugar, cherry, malt, roasted, pepper	350 µg/kg
Safranal	17.64 ± 1.06 ^a^	11.05 ± 2.39 ^b^		
*Total unsaturated aldehydes*	*569 ± 67 ^a^*	*577 ± 180 ^a^*		
(*E*,*E*)-2,4-Decadienal	280 ± 36 ^a^	309 ± 125 ^a^	Baked, grease and oil	0.07 µg/kg
2-Heptenal	55.67 ± 7.78 ^a^	69.11 ± 5.27 ^a^	Grease and fruity	13 µg/kg
2-Octenal	64.77 ± 17.35 ^a^	54.50 ± 12.91 ^a^	Roasted pea nuts, fatty	3 µg/kg
2-Nonenal	65.27 ± 13.06 ^a^	45.75 ± 23.94 ^a^	Fatty, pungent	0.69 µg/L
(*E*,*Z*)-2,4-Decadienal	38.70 ± 1.76 ^a^	42.40 ± 18.49 ^a^	Fatty, cooked grain, deep-fried	0.07 µg/kg
4-Heptenal	36.08 ± 1.96 ^a^	21.84 ± 2.32 ^b^	Fishy, fish-oil-like	0.025 µg/L
2-Decenal	16.10 ± 1.70 ^a^	21.67 ± 7.54 ^a^		
2-Butyl-2-octenal	11.87 ± 4.96 ^a^	12.31 ± 3.42 ^a^		
*Total branched aldehydes*	*727 ± 129 ^a^*	*325 ± 101 ^b^*		
3-Methylbutanal	446 ± 71 ^a^	180 ± 54 ^b^	Aldehydic, ethereal, acrid, almond, chocolate, malty, pungent	0.5 µg/L
2-Methylbutanal	281 ± 58 ^a^	145 ± 48 ^b^	Chocolate, musty, nutty, malty, almond, fermented	1.5 µg/L
2-Methylpropanal	73.35 ± 17.56 ^a^	32.00 ± 12.46 ^b^	Aldehydic, caramel, cocoa, malt, nut	
Total furan derivatives	6633 ± 323 ^a^	5174 ± 910 ^b^		
2-Pentylfuran	5601 ± 339 ^a^	4222 ± 836 ^b^	Fruity, green, earthy, bean, buttery, fishy, grassy	6 µg/kg
Furfural	483 ± 12 ^a^	470 ± 26 ^a^	Almond, baked potatoes, bread, burnt, spice, bready	0.002–0.713 ppm
2-Ethylfuran	300 ± 13 ^a^	254 ± 19 ^b^		
2-Butylfuran	115 ± 17 ^a^	113 ± 30 ^a^		
2-Methylfuran	83.22 ± 5.48 ^a^	74.61 ± 13.64 ^b^	Sweet, green, fruity	
3-Methylfuran	50.72 ± 0.26 ^a^	38.90 ± 5.53 ^b^		
Total benzene derivatives	2402 ± 147 ^a^	2538 ± 443 ^a^		
Toluene	1764 ± 72 ^a^	1932 ± 308 ^a^	Sweet, pungent, benzene-like	0.33 ppm
*m*-Xylene	218 ± 42 ^a^	226 ± 48 ^a^		0.041 ppm
*o*-Xylene	182 ± 25 ^a^	185 ± 45 ^a^	Sweet	0.38 ppm
Ethylbenzene	81.43 ± 7.98 ^a^	70.39 ± 18.93 ^a^	Gasoline	0.2 mg/kg
1,2,3-Trimethylbenzene	87.88 ± 7.27 ^a^	57.05 ± 7.63 ^b^	Aromatic	0.006–2.4 ppm
*p*-Xylene	69.96 ± 6.15 ^a^	68.06 ± 19.50 ^a^	Cold meat fat, metal	0.058 ppm
Total alcohols	1280 ± 191 ^a^	1260 ± 280 ^a^		
*Total aliphatic alcohols*	*797 ± 115 ^a^*	*920 ± 188 ^a^*		
1-Octene-3-ol	267 ± 45 ^a^	259 ± 59 ^a^	Earthy, fishy, fat, mould, mushroom	1 µg/kg
1-Hexanol	194 ± 8 ^b^	285 ± 35 ^a^	Herbal, flower, fruit, green, wood	0.7 mg/kg
2-Ethyl-1-hexanol	177 ± 53 ^a^	185 ± 59 ^a^	Citrus, green, flowery	
1-Pentanol	92.91 ± 2.14 ^a^	99.32 ± 14.96 ^a^	Fermented, oily, sweet, vinegar	5.0 mg/kg
1-Heptanol	41.12 ± 6.26 ^a^	58.79 ± 12.31 ^a^		0.2 mg/kg
1-Octanol	24.91 ± 6.85 ^a^	33.57 ± 9.86 ^a^	Fatty, waxy	120 µg/kg
*Total phenols*	*483 ± 76 ^a^*	*340 ± 96 ^a^*		
2,4-Di-*tert*-butylphenol	187 ± 24 ^a^	136 ± 37 ^a^		
2-Methoxy-4-vinylphenol	132 ± 47 ^a^	84.75 ± 51.56 ^a^	Clove-like, smoky	19 µg/L
4-Vinylphenol	99.81 ± 17.48 ^a^	83.86 ± 31.50 ^a^		
Phenol	46.60 ± 5.49 ^a^	11.78 ± 1.51 ^b^	Tarry	0.0045–1.95 ppm
4-(1-Methylpropyl) phenol	16.99 ± 1.68 ^a^	23.33 ± 5.86 ^a^		
Total ketones	1119 ± 39 ^a^	718 ± 101 ^b^		
2-Heptanone	462 ± 7 ^a^	246 ± 28 ^b^	Cheesy, fruity, spicy, sweet	0.14 mg/kg
2-Nonanone	201 ± 21 ^a^	144 ± 27 ^b^	Fragrant, fruit, green, hot milk, cheese, coconut	0.08 mg/kg
2,3-Pentanedione	91.01 ± 8.55 ^a^	74.37 ± 8.12 ^b^	Buttery	
6-Methyl-5-heptane-2-one	103 ± 12 ^a^	54 ± 1 ^b^		
2-Octanone	78.20 ± 6.53 ^a^	46.90 ± 9.07 ^b^		0.04 mg/kg
2-Undecanone	62.09 ± 20.66 ^a^	49.73 ± 22.03 ^a^	Orange, grassy, fresh	0.08 mg/kg
2-Propanone	56.03 ± 8.57 ^a^	40.94 ± 7.31 ^b^		
2,3-Octanedione	33.29 ± 7.69 ^a^	30.80 ± 5.83 ^a^	Dill and earthy	
3-Octene-2-one	31.81 ± 0.65 ^a^	32.38 ± 3.88 ^a^	Rose	
Total pyrazines and pyrroles	1558 ± 34 ^a^	590 ± 23 ^b^		
2,6-Dimethylpyrazine	530 ± 26 ^a^	174 ± 11 ^b^	Cooked meat	1720 ng/L air
2-Methylpyrazine	361 ± 13 ^a^	135 ± 1 ^b^		
2,5-Dimethyl-3-ethylpyrazine	326 ± 21 ^a^	126 ± 16 ^b^		3.6 ng/L air
2,5-Dimethylpyrazine	191 ± 13 ^a^	90 ± 6 ^b^	Cocoa, roast beef, roasted nut, burnt, chocolate	1820 ng/L air
Pyrazine	70.70 ± 11.08 ^a^	37.00 ± 2.73 ^b^	Bitter taste	0.16 mg/kg
2-Ethyl-5-methylpyrazine	52.65 ± 4.97 ^a^	15.43 ± 2.38 ^b^		
2-Formylpyrrole	26.13 ± 4.94 ^a^	11.31 ± 2.23 ^b^		
Total aliphatic hydrocarbons	1104 ± 77 ^a^	440 ± 104 ^b^		
3-Ethyl-2-methyl-1,3-hexadiene	974 ± 50 ^a^	324 ± 60 ^b^		
Pentane	76.52 ± 27.01 ^a^	57.25 ± 30.67 ^a^	Sweet	1.29–1147 ppm
Octane	38.80 ± 3.06 ^a^	41.42 ± 8.21 ^a^	Gasoline, alkane	0.66–235 ppm
Heptane	14.55 ± 2.53 ^a^	16.74 ± 6.08 ^a^		
Total terpenes	536 ± 52 ^a^	46 ± 21 ^b^		
*Total sesquiterpenes*	*502 ± 46* ^a^	*20 ± 16* ^b^		
Ar-curcumene	233 ± 23 ^a^	20.11 ± 15.87 ^b^		
a-Zingiberene	203 ± 16	n.d		
b-Sesquiphellandrene	55.91 ± 6.38	n.d		
b-Bisabolene	10.18 ± 3.29	n.d		
*Total monoterpene ketones*	*33.0 ± 6.2 ^a^*	*26.2 ± 6.9 ^a^*		
Geranyl acetone	33.01 ± 6.19 ^a^	26.18 ± 6.88 ^a^		
Total lactones	101 ± 7 ^a^	119 ± 25 ^a^		
g-Nonalactone	101 ± 7 ^a^	119 ± 25 ^a^		
Total sulfur compounds	144 ± 16 ^a^	48.6 ± 7.2 ^b^		
Dimethyl disulfide	86.16 ± 5.32 ^a^	17.26 ± 2.14 ^b^	Garlic, putrid, asparagus	0.0022 ppm
Dimethyl sulfide	57.56 ± 10.64 ^a^	31.34 ± 5.09 ^b^	Sulfurous, onion, sweet	0.0030 ppm
Total acids	12.7 ± 5.2 ^a^	7.2 ± 0.7 ^a^		
Nonanoic acid	12.68 ± 5.02 ^a^	7.18 ± 0.70 ^a^		
Total volatiles	23450 ± 801 ^a^	18210 ± 2518 ^b^		

The descriptions of sensory attributes and detection thresholds were adopted from [12,23,24,25,26,27,28,29,30,31,32,33,34,35,36]. The major classes of aroma compounds identified are presented in italics.

## Data Availability

The original contributions presented in this study are included in the article/Appendix A. Further inquiries can be directed to the corresponding author.

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
