# Peer review of "Physico-Chemical and Sensory Characteristics of Extruded Cereal Composite Flour Porridge Enriched with House Crickets (Acheta domesticus)"

_foods, 2025, doi:10.3390/foods14162893_

Round 1

Reviewer 1 Report

Comments and Suggestions for Authors

C1.

Line 162

I believe that section „Aroma profile“ should be presented as subheading with number 2.3.1

C2.

Line 169

I believe that section „Analysis of volatile compounds“ should be presented as subheading with number 2.3.2

C3.

Line 233

I kindly ask the authors to provide reference or to explain why they have used this recipe for sensory analysis. The recipe is the following „To make porridge for sensory tests, 50 g of flour were mixed with 235 mL of hot boiling water (1:5) and 15 g of sugar were added. The flour was continuously stirred for 5 min and kept in a vacuum flask for serving.“

In addition, please provide the temperature of served samples to panelists. This is particularly important since it is known how temperature of sample can affect sensory acceptance of various food samples.

C4.

Line 233

There is a question regarding familiarity of sensory panel with the type of porridge. Please answer the question were the participant regular consumers of this type of porridge or it was introduced to them for the first time.

C5.

Line 260

In the following sentence „Sensory acceptability and physico-chemical measurement data were analyzed using JMP Pro software, version 17.0 at a 5 % significance level.“ Please indicate the producer and producer country of the software in the brackets.

C6.

Line 272-273

In table 1. ± needs explanation. I believe that this is a standard deviation of measurements. However, please indicate this in the table caption and within the table.

In addition, if this is standard deviation, I kindly ask the authors to check their statistical data once again since standard deviation seems to be very low. Konika Minolta often gives results with much higher standard deviation.

C7.

Line 267

In my opinion, it would be beneficial if authors could provide images of samples in the manuscript or at least in supplementary files.

C8.

Line 268

When discussing color parameters, it is important to know how long samples were stored, in which type of packaging and at which temperature before analysis. Here, it would be interesting to know whether or not color parameters change and to what extent during storage of the samples.

C9.

Line 300 – 301

In table 2, please explain the meaning of the ± symbol. I believe that this represents standard deviation like in table 1. However, this information should be included in table caption and within the table.

C10.

Line 489 -545

Please check if the pages 18 and 19 should be in landscape or portrait.

C11.

Line 487

In figure 1, please indicate the meaning of vertical bars. Again, I believe they represent standard deviation. In any case, include this into the figure caption.

C12.

Line 616

Similar to figure 1, please indicate what is the meaning of vertical bars.

In addition, check this in figure S2.

Author Response

Dear reviewer 1,

Thanks for your comments, questions and advise regarding our manuscript. The response file is hereby attached with changes indicated by red-colored font.

Reviewer 2 Report

Comments and Suggestions for Authors

Porridge is a healthy food, and when enriched with crickets, it can improve its nutritional value. However, the paper has many mistakes, as given below:

Observations:

The abstract should be rewritten in a crisp and concise form

Keywords (Acheta domesticus, sensory characteristics, physico-chemical characteristics, rheological properties, extrusion, pasting properties): Keywords closely repeat the title; consider revising them for uniqueness and broader indexing impact.

Line 42: Mention ‘Nutritional quality’. There is no nutritional analysis that was conducted. Please justify and validate.

Line 70-71: Please elaborate on other foods on the nutritious composite incorporated for nutrient enhancement, supported with appropriate citations.

Lines 88-90: You mention an increase or decrease of nutrients. Please justify with a proper reason and explain how much it increases or decreases, providing a proper citation.

Line 91-92: It looks like the Last paragraph of the introduction. You should rewrite and align with the previous paragraph. Again, Citation is missing in lines 94-98.

The overall observation in the introduction is that you should rewrite your abstract and narrate a story that fulfills your research objective. If you state anything that has already been reported, you should cite the relevant source in your article.

Last paragraph of your intro. Physicochemical analysis. There is no physico-chemical analysis. Either you should remove it or incorporate that result.

In sec. Materials and methods

Section 2.1 Preparation of Flour Sample: This section is too long. Unnecessary information is there. Please keep it concise.

Line no 128: Mention specific temp. (No need to mention ‘Room Temperature’)

Line no 136: Please complete the software details. The origin and year are missing.

The flour containing house crickets was formulated to meet the protein and micronutrient requirements of lactating women. It is reported that for a solid food to be considered a source of protein and micronutrients, it should fulfil at least 10 % and 15 % of the age-specific protein and micronutrient (vitamins and minerals) requirements, respectively.’ This looks like an introductory part. You should remove it from the M&M sec.

Convert this section into 2 segments: 1) Sample procurement and preparation; in this section, you mention how you prepare your samples. 2) Extrusion processing. In this section, you mention extrusion processing.

Line no 151: How did you adjust the moisture? Please justify and elaborate.

Line no 157: Write proper M&M for colour.

Line no 160: Sec. Colour- Write the formula for (ΔE). As well as (ΔC).

Line no 163: A sample (15 g ± 0.1 %) and Line no 166: A 5 g (± 0.1 %). It should be in uniform throughout the manuscript. Either use brackets or omit them, but the style should be uniform.

Line no. 169: Analysis of volatile compounds - Was this method developed by you or sourced elsewhere? Please specify.

Or, if you took information from any database, cite and reference it.

Rheological properties of porridges- Citations are missing.

Line no 176: Thermostatic agitator- Write instrument detail

Line no 216: Water bath – Mention its specification.

For sensory analysis: Which type of water do you take for sample preparation?

Mention sensory conditions. Timing, room condition, and what was the sensory approach? 

Line No 249-252: No need to mention here repetitively. You already mentioned in S1 (questionnaire)

For Table 1, please verify the statistics for the values.

Line no 274: mention about (p value), whatever it is.

Line no 282: Explain the scientific reason. Why is it a dark appearance?

Line No 310: Write correctly and statistically the line ‘Significantly higher….’.

In the result and discussion, you mentioned significantly higher…/ Write in proper scientific language. And use statistical terms like (p > 0.05) and 'significantly higher,' specifying the exact amount in brackets.

In Figure 1, mention statistical analysis in the figure with superscripts.

Line no 496-497: ‘The control flour (140C) had a higher starch content, 498 since house cricket substitution is not expected to contribute to starch in 140U.’ How much starch is there? What is the impact of extrusion? And there is no analysis regarding starch quantification.

Explain the significance of Viscosity analysis in your study.

Figure 2: It is not clear. Revised it and made it clear and visible

Line no 577: Figure 3, Double comma is there. Make it correct.

In conclusion, mention future scope and limitations.

Author Response

Dear reviewer 2,

Thanks for the important suggestions, comments and questions that have enabled us to improve the quality of our manuscript. The report indicating how each comment was addressed is hereby attached.

Round 2

Reviewer 2 Report

Comments and Suggestions for Authors

NA